# Incidence of Screw Loosening in Cortical Bone Trajectory Fixation Technique between Single- and Dual-Threaded Screws

**DOI:** 10.3390/medicines8090050

**Published:** 2021-09-02

**Authors:** Chao-Hsuan Chen, Chih-Hsiu Tu, Der-Cherng Chen, Hsiang-Ming Huang, Hao-Yu Chuang, Der-Yang Cho, Da-Tian Bau, Han-Chung Lee

**Affiliations:** 1Graduate Institute of Biomedical Sciences, China Medical University, Taichung 40402, Taiwan; jemileiia@yahoo.com.tw; 2Department of Neurosurgery, China Medical University Hospital, Taichung 40447, Taiwan; eddietu.tw@gmail.com (C.-H.T.); vincenchen1966@yahoo.com.tw (D.-C.C.); d5057@mail.cmuh.org.tw (D.-Y.C.); 3Department of Surgery, School of Medicine, China Medical University, Taichung 40402, Taiwan; 4Department of Neurosurgery, China Medical University Hsinchu Hospital, Hsinchu 30272, Taiwan; d10190168@gmail.com; 5Department of Neurosurgery, Tainan Municipal An-Nan Hospital-China Medical University, Tainan 70965, Taiwan; greeberg1975@gmail.com; 6Terry Fox Cancer Research Laboratory, Department of Medical Research, China Medical University Hospital, Taichung 40402, Taiwan; 7Department of Bioinformatics and Medical Engineering, Asia University, Taichung 41354, Taiwan; 8Department of Neurosurgery, Everan Hospital, Taichung 41159, Taiwan

**Keywords:** cortical bone trajectory, midline lumbar inter-body fusion, screws loosening, screws thread, fixation length

## Abstract

**Purpose:** This study aims to elucidate the radiological outcome after Cortical bone trajectory (CBT) screw fixation and whether dual-threaded (DT) screws should be used in the fusion surgery. **Methods:** 159 patients with degenerative lumbar disorder who had undergone midline lumbar inter-body fusion surgery by CBT screw-fixation technique (2014 to 2018). Patient subgroups were based on single-threaded (ST) or DT screw, fixation length, as well as whether fixation involved to sacrum level (S1). Serial dynamic plain films were reviewed and an appearance of a halo phenomenon between screw–bone interfaces was identified as a case of screw loosening. **Results:** 29 patients (39.7%) in ST group and 10 patients (11.6%) in DT group demonstrated a halo phenomenon (*p* < 0.0001 ****). After subgrouping with fixation length, the incidence rates of a halo phenomenon in each group were 11.1%:3% (ST-1L vs. DT-1L), 37%:13.8% (ST-2L vs. DT-2L), and 84.2%:23.5% (ST-3L vs. DT-3L). Among the 85 patients with a fixation involved in S1, 26 patients (52%) with single-threaded screw (STS group) and 8 patients (22.8%) with dual-threaded screw (DTS group) demonstrated a halo appearance (*p* = 0.0078 **). After subgrouping the fixation level, the incidence of a halo appearance in each group was 25%:0% (STS-1L vs. DTS-1L), 40.9%:26.3% (STS-2L vs. DTS-2L), and 87.5%: 30% (STS-3L vs. DTS-3L). **Conclusion:** Both fixation length and whether fixation involved to S1 contribute to the incidence of screw loosening, the data supports clinical evidence that DT screws had greater fixation strength with an increased fixative stability and lower incidence of screw loosening in CBT screw fixation compared with ST screws. **Level of evidence:** 2.

## 1. Introduction

Spinal fusion is currently the standard surgical treatment for various lumbar spinal disorders. Pedicle screw fixation along the axis of the pedicle of the lumbar arch is the most common and reliable procedure in fusion surgery. With this entry point and route, the screw contacts the cancellous bone of the pedicle and vertebral body instead of the cortical bone of the pedicle. However, in the aging society, resorption of the cancellous bone trabeculae in osteoporosis patients who underwent pedicle screw fixation may present instrument complications, such as screw loosening or displacement [1,2]. Cortical bone trajectory (CBT) screw insertion is a novel fusion technique in spinal surgery that was first described by Santoni et al. in 2009; such a technique increased the uniaxial yield pullout load by 30%, compared with traditional pedicle screws (TPS) [3]. Given the differences in entry point and trajectory compared with the conventional pathway, this technique minimizes the engagement of trabecular bone within the pedicle and offers greater amount of screw-cortical bone interception. Several biomechanical studies have demonstrated that the insertion torque and mechanical properties of CBT screws were equivalent to or greater than the conventional trajectory and provided similar or better fusion stability [4,5]. However, TPS method uses cancellous screw, whereas CBT uses cortical screws, which maximize the thread contact with the high-density cortical bone. In 2015, Ueno et al. reported that the specific trajectory itself has a major impact on the increased strength obtained with CBT in a biomechanical study [6]. However, whether differently designed thread screws are suitable for CBT in clinical use remains unclear.

To the best of our knowledge, no reports have examined the incidence of screw loosening in CBT with differently designed thread screws. Therefore, we retrospectively reviewed 159 patients with degenerative lumbar disorder and who underwent midline lumbar inter-body fusion (MIDLF) surgery by CBT screw-fixation technique with single-(ST) or dual-threaded (DT) screws from November 2014 to May 2018 and followed-up for more than 6 months. This study aims to elucidate the radiological outcome after CBT screw fixation and whether DT screws should be used in such surgery.

## 2. Methods

### 2.1. Patient Selection and Characteristics

Since November 2014, we began performing MIDLF with CBT fixation in our institute for degenerative spinal disorders because of the excellent fixation strength declared by Santoni et al. [3]. To date, 159 patients who had consecutively undergone CBT screw fixation were followed-up for more than 6 months. All the patients exhibited degenerative spinal disorders, including grade one to grade two spondylolisthesis, disc degeneration with vacuum appearance on dynamic plain films, which caused motion back pain and symptoms of nerve entrapment. Patients with simple lumbar, stenosis caused by herniated intervertebral disc (HIVD) or hypertrophic flavum ligament, were treated with decompression alone and not included in this study. Besides, patients with traumatic vertebral column fracture, infection, or tumor invasion were also excluded. A total of 62 males and 97 females were included in this study. The mean age at the time of surgery was 59.12 years (range: 26–87 years). The mean duration of the follow-up was 13.41 months (range: 6–33 months) (Table 1).

A total of 73 patients (36 males and 37 females with mean age of 56.97 years) who underwent CBT fixation with ST screws were classified into the ST group, and a total of 86 patients (with 31 males and 55 females with mean age of 62.12 years) with DT screws were classified into the DT group (Table 2). Neither patients nor surgeons realized the difference of fixation strength between ST screw and DT screw because there were no reports which examined the incidence of screw loosening in CBT with differently designed thread screws. All the patients included in this study were well-educated about post-operation self-cares including brace wearing, posture maintenance, and the avoidance of heavy lifting. Subgroups were made in each group based on the numbers of fixation levels; for example, subgroup ST-1L means a patient with level 1 ST screw fixation, and subgroup DT-1L means a patient with level 1 DT screw fixation. The study was approved by Research Ethics committee China Medical University and hospital on Aug. 09, 2019 (CMUH108-REC2-113). 

### 2.2. Instrument Characteristics

Until now, the National Health Insurance Administration of our country only offers ST screws for patients who need instrumentation. Therefore, patients must pay at their own expense or private insurance supported for using DT screws. We used both ST and DT screws in patients who underwent CBT surgery based on patients’ insurance situation and agreement of self-funding. The ST screw is a traditional cancellous screw used in conventional surgery with a constant pitch of 2 mm (Figure 1A). The DT screw we used was a cannulated screw with a cortical thread region of 1.5 mm in pitch and a cancellous thread region with a 3 mm crest-to-crest distance (Figure 1B). The screws we inserted were all 5.5 mm in diameter and 40 mm in length in all the levels of vertebrae except L5, which is 5.5 mm in diameter and 35 mm in length.

### 2.3. Surgical Procedure

Instead of a traditional mini-open MIDLF with CBT, we chose the minimally invasive MIDLF surgical procedure reported by Chen et al. [7] to achieve smaller wound incision, less muscular damage, and less exposure to radiation. The wound incision was made between the entry points of the fusion level with minimum muscular dissection to the medial pars interarticularis. The guiding pins were inserted under anteroposterior view in fluoroscopy followed by mediolateral and caudocephalad directions. However, the entry point of the caudal level was on the articular surface of the superior articular process, and the trajectory took a mediolateral path parallel to the endplate to minimize caudal muscular dissection. Then, decompression with cage interbody fusion was performed using a microscope. Finally, the screws were inserted through the pilot tract or guide pins.

### 2.4. Radiological Evaluations

Based on the research by Sandén in 2004, a radiolucent zone is a good indicator of the loosening of a pedicle screw [8]. In the present study, either a halo sign at the screw–bone interface on anteroposterior plain films or an evidence of screw motion in the dynamic images was defined as screw loosening (Figure 2). All the patients involved in this study were followed with anteroposterior and dynamic lateral plain films at at least 1, 2, 3, and 6 months post-operation. The serial images were reviewed by two neurosurgeons familiar with CBT technique. Cases were excluded when two neurosurgeons did not agree on the results simultaneously.

### 2.5. Statistical Analysis

Data-plotting and analysis were performed using GraphPad Prism. Data were represented as mean ± SD. All categorical data are presented as a percentage or number. The results among groups were compared using unpaired *t* test, and Fisher’s exact probability test. Significance was set at *p* < 0.05 *, *p* < 0.01 **, *p* < 0.001 ***, *p* < 0.0001 ****.

## 3. Results

Starting in November 2014, we began performing MIDLF with CBT fixation in the Institute for Degenerative Spinal Disorders. To date, 159 consecutive patients have undergone CBT screw fixation and had follow-ups for more than 6 months. Although older age manifested in the DT group and patients in this group were considered to have a worse median bone quality and present a higher incidence of screw loosening (*p* = 0.0055 **), only 10 out of 86 (11.6%) patients in DT group exhibited halo sign or screw motion on the plain films (Fisher’s exact test, *p* < 0.0001 ****) in contrast to 29 out of 73 (39.7%) patients in ST group and (Figure 3). Patients in group ST and DT were classified into subgroups ST-1L, ST-2L, and ST-3L and DT-1L, DT-2L, and DT-3L, which represent the fixation lengths of 1, 2, and 3 levels. We found that the incidences of halo phenomenon in subgroups ST-1L vs. DT-1L, ST-2L vs. DT-2L, and ST-3L vs. DT-3L were 11.1%:3% (Fisher’s exact test, *p* = 0.3179), 37%:13.8% (Fisher’s exact test, *p* = 0.0410 *), and 84.2%:23.5% (Fisher’s exact test, *p* = 0.0006 ***), respectively (Figure 4), suggesting the DT group can reduce the incidence rates of a halo phenomenon, resulting from the greater of fixation length. Among 85 patients have fixation lengths involved in the sacrum level (S1), 26 patients out of 50 patients (52%) with ST screw and 8 out of 35 patients (22.8%) with DT screw exhibited the halo phenomenon. They were labeled as groups STS and DTS (Fisher’s exact test, *p* = 0.0078 **) (Figure 3). They were also classified into subgroups STS-1L, STS-2L, and STS-3L and DTS-1L, DTS-2L, and DTS-3L according to the fixation length. The incidences of the halo phenomenon in subgroups STS-1L vs. DTS-1L, STS-2L vs. DTS-2L, and STS-3L vs. DTS-3L were 25%:0% (Fisher’s exact test, *p* = 0.5147), 40.9%:26.3% (Fisher’s exact test, *p* = 0.5100), and 87.5%:30% (Fisher’s exact test, *p* = 0.0085 **), respectively (Figure 5), indicating whether fixation involved to scrum level also contribute to the incidence of screw loosening.

## 4. Discussion

Decompression with instrumented spinal fusion has become a common and reliable technique for treating a variety of spinal disorders. Many approaches have been advocated in the past decades, and posterior approaches are more familiar to most spine surgeons. Traditional pedicle screw fixation with lumbar interbody fusion usually requires longer incision and wide dissection of the muscular structure, which may cause damage to the posteromedial nerve branch crossing the facet joints and in turn lead to prolonged hospitalization and chronic low-back pain. CBT screw insertion is a novel fusion technique in spinal surgery that was first described by Santoni et al. in 2009; such a technique increased the uniaxial yield pullout load by 30% compared to TPS [3]. Matsukawa et al. showed that the screw insertion torque by CBT method was 1.71 times higher than that by the traditional technique [4]. Several biomechanical studies have also advocated the favorable results using CBT screw fixation in cadaveric lumbar specimens [5]. Mai et al. suggested that the potential advantages from the CBT screw, such as screw purchase may increase linearly with age and in osteoporotic patients [9]. Given the possibility of eliminating the disadvantages of the traditional pedicle trajectory and the better biomechanical results, CBT screw-fixation technique has become increasingly popular worldwide in recent years.

Surgeons often use cancellous screw for TPS method. However, Dr. Santoni used cortical screws for CBT to maximize thread contact with the high-density cortical bone. In 2015, Ueno et al. reported that the specific trajectory itself has a major impact on the increased strength obtained by CBT compared with the differently designed thread screws [6]. However, whether these differently designed thread screws are suitable for CBT in clinical use remains unclear. To the best of our knowledge, no clinical study has been conducted on the incidence of screw loosening in CBT with differently designed thread screws in vivo.

Previous experimental studies by Mckinley [10] and Chen [11] suggested that the screw should be sufficiently inserted into the vertebral body for effective vertebrae loading. However, in the original study of CBT reported by Santoni et al., the screw did not extend into the middle column of the vertebrae [3]. Matsukawa et al. revealed that the cephalad angle and screw length within the lamina had a major influence on the fixation strength of CBT. Therefore, they suggested using a more caudal entry point with the trajectory passing the lower part of the pedicle to reach the middle column of the vertebrae [12]. Through this pathway, half of the screw will be involved in the pedicle and will contact a large volume of cortical bone [13], whereas the other half will be in the vertebral body and enrolled with abundant cancellous bone. Therefore, in the pedicle part of the DT screw, the dense thread maximizes the contact with the cortical bone. On the other hand, the part in the vertebrae, the loose thread has a deeper cut and wider space enable to hold a large bone volume between the threads.

In the present study, patients who underwent MIDLF with CBT fixation by using ST screws had more than three times of incidence of screw loosening than those who used DT screws (39.7% to 11.4%). Although the incidences both increased with fixation length (11.1% to 3% at 1 level, 37% to 13.8% at 2 level, and 84.2% to 23.5% at 3 level), the difference became more significant when the fixation length was beyond level 3. Moreover, nearly all of the screws loosened at S1 level with the following incidences: STS-1L: 25% (3 in 12), DTS-1L: 0% (0 in 6), STS-2L: 40.9% (9in 22), DTS-2L: 26.3% (5 in 19), STS-3L: 87.5% (14 in 16), and DTS-3L: 30% (3 in 10). Only one patient of in the 3-level fixation with ST screw had bilateral S1 and one L3 screw loosening. The lumbosacral junction remains difficult to achieve for successful spinal fusion, and the prevention of construct failures, such as screw loosening at the S1 [14], remains challenging. Excessive mechanical stress on the lumbosacral junction, abundant cancellous bone from the first sacral vertebral body to the sacral ala, and lack of true pedicle may give rise to this phenomenon [15]. Sakaura et al. reported 193 patients who had consecutively undergone single-level fusion by CBT with 46.2% screw-loosening rate at S1 [16]. In the present study, this incidence is similar to that in group STS (52%), but significantly higher than that in group DTS (22.8%).

## 5. Conclusions

Besides to fixation length and fixation involved to scrum level contribute to the incidence of screw loosening. The data supports the evidence that the DT screws had greater fixation strength, with an increase in the fixative stability and a lower incidence of screw loosening in CBT screw fixation compared with ST screws. Therefore, we strongly suggest that surgeons use DT screws when performing CBT fixation surgery beyond three levels or at S1in order to reduce these incidences of screw loosening.

## Figures and Tables

**Figure 1 medicines-08-00050-f001:**
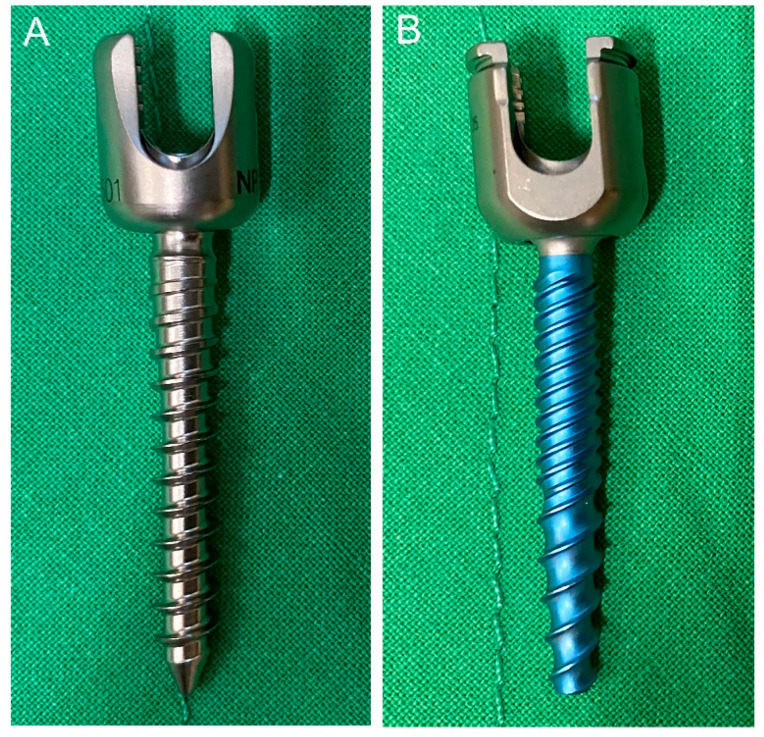
(**A**) The single-threaded screw with constant pitch of 2 mm. (**B**) The dual-threaded screw was cannulated screw with a cortical thread region of 1.5 mm in pitch and a cancellous thread region with a 3 mm crest-to-crest distance.

**Figure 2 medicines-08-00050-f002:**
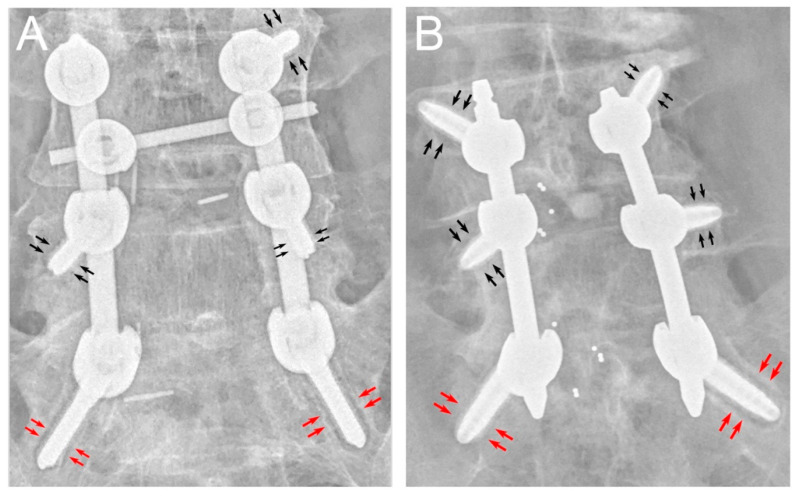
Plain radiograph of (**A**) ST and (**B**) DT CBT screws. Red arrows represent halo phenomenon around the CBT screws and black arrows represent CBT screws without halo phenomenon.

**Figure 3 medicines-08-00050-f003:**
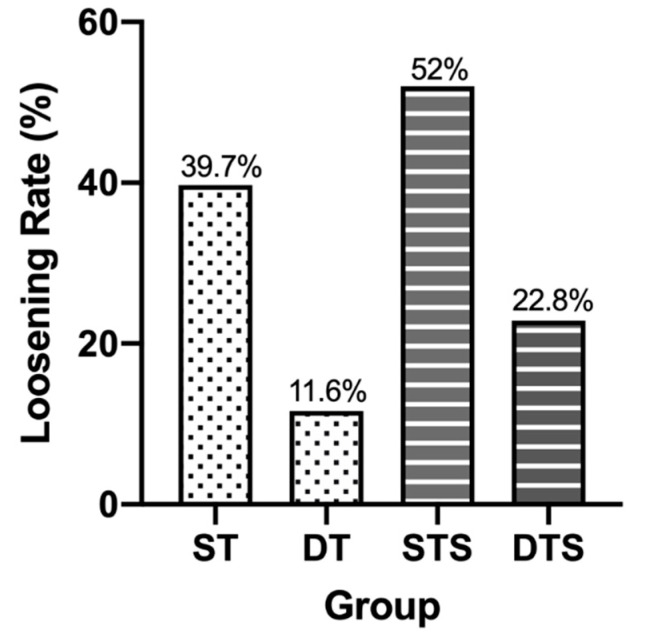
Patients in ST group (single-threaded screw) had 39.7% while only 11.6% patients in DT group (dual-threaded screw) has halo sign appearance or screw motion on plain films (Fisher’s exact test, *p* < 0.0001). With involvement to sacral level, 52% patients with single-threaded screws (STS group) and 22.8%patients with dual-threaded screws (DTS group) exhibited the halo phenomenon (Fisher’s exact test, *p* = 0.0078).

**Figure 4 medicines-08-00050-f004:**
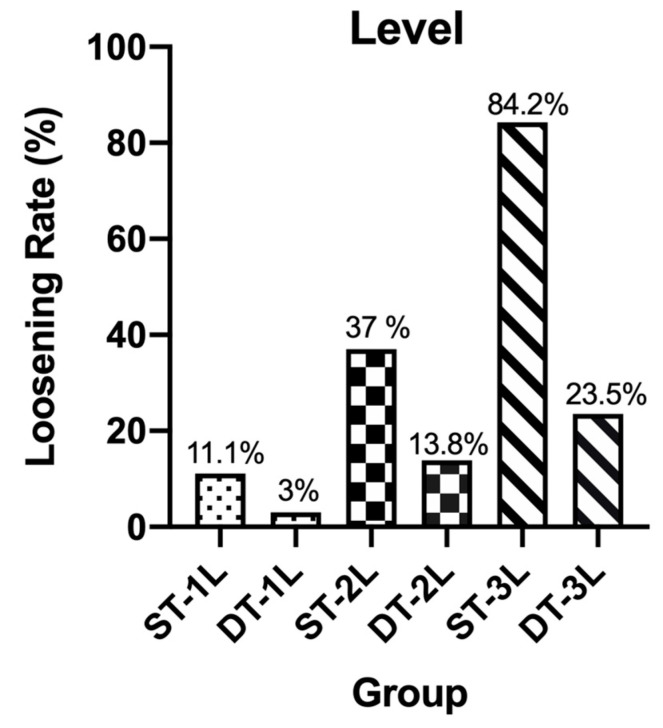
Subgroups ST-1L, DT-1L, ST-2L, DT-2L, ST-3L and DT-3L, which represent the fixation lengths of 1, 2, and 3 levels. We found that the incidences of halo phenomenon in subgroups ST-1L vs. DT-1L, ST-2L vs. DT-2L, and ST-3L vs. DT-3L were 11.1%:3% (Fisher’s exact test, *p* = 0.3179), 37%:13.8% (Fisher’s exact test, *p* = 0.0410), and 84.2%:23.5% (Fisher’s exact test, *p* = 0.0006), respectively.

**Figure 5 medicines-08-00050-f005:**
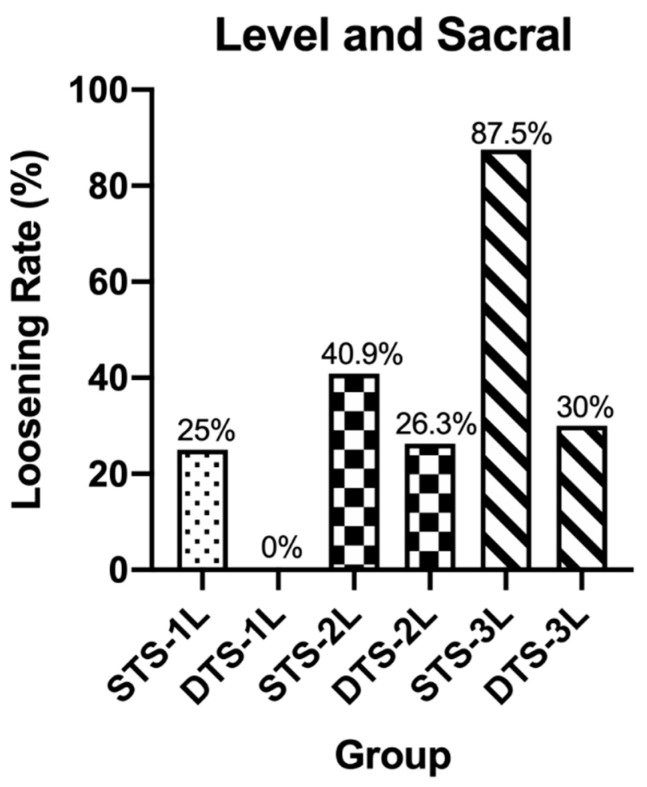
In patients with sacral involvement, the incidences of halo phenomenon in subgroups STS-1L vs. DTS-1L, STS-2L vs. DTS-2L, and STS-3L vs. DTS-3L were 25%:0% (Fisher’s exact test, *p* = 0.5147), 40.9%:26.3% (Fisher’s exact test, *p* = 0.5100), and 87.5%:30% (Fisher’s exact test, *p* = 0.0085), respectively.

**Table 1 medicines-08-00050-t001:** Patient’s demographics.

Characteristics	N = 159	
Age, year		
	Range	26–87
	Mean ± SD	59.74 ± 13.31
Gender, No. (%)		
	Male	62(40.12%)
	Female	97(59.88%)
Follow-up period (Month)		
	Range	5–39
	Mean ± SD	13.75 ± 8.14

**Table 2 medicines-08-00050-t002:** Characteristics of Group A and B.

Characteristics	N = 159				
	Screw				
	A. Single-threaded (*n* = 73)		B. Dual-threaded (*n* = 86)		*p*-value
Age, year	Range	27–87	Range	26–85	0.0055 **
	Mean ± SD	56.82 ± 12.33	Mean ± SD	62.21 ± 13.69	
Gender, No. (%)	Male	36 (49.32%)	Male	31 (36.05%)	
	Female	37 (50.68%)	Female	55 (63.95%)	

** Significance was set at *p* < 0.01.

## Data Availability

The datasets generated during and analyzed during the current study are available from the first corresponding author on reasonable request.

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
