# Peer review of "Incidence of Screw Loosening in Cortical Bone Trajectory Fixation Technique between Single- and Dual-Threaded Screws"

_medicines, 2021, doi:10.3390/medicines8090050_

Round 1
Reviewer 1 Report
The paper is a clinical study on the Incidence of Screw Loosening in Cortical Bone Trajectory Fixation Technique between Single- and Dual- threaded Screws Running head by CBT .
The authors made a great work in terms of methodology and the paper sounds scientific and well written.
However, some improvements are mandatory before acceptance.

Author Response
We greatly appreciate the efforts of the Editorial Board and the reviewers for giving us valuable comments on our manuscript. As requested, we enclosed a revised version of our manuscript in response to the extensive and insightful reviewer’s comment. Included below is our point-by-point response to the comment.
1.In the introduction, and sometimes in the text:
“displacement(2, 9).” Please fix the references style in all the text, following MDPI Journals style.
- As reviewer’s suggestion, we correct the references by MDPI Journal style.
2.In methods section:
- “Until now, the National Health Insurance Administration of our country only offers 90 ST screws for patients who need instrumentation. Therefore, patients have to pay at their 91 own expense or private insurance supported for using DT screw. We used both ST and 92 DT screws in patients who underwent CBT surgery based on patients’ insurance situation 93 and agreement of self-paying.” Is this consideration necessary, or can change the evaluation of the evidence contained in the article?
- Please underline what type of study was conducted.
Although the fact that insurance differences may discriminate the people and that would be affect the results, we still think that have to be disclosed. Because neither patients nor surgeons realized the difference of the fixation strength between ST and DT screw because there were no reports have verified the clinical outcomes in CBT with differently designed thread screws. Moreover, the initial idea we used DT screws is due to the cannulated design, which can reduce the radiation exposure of operative surgeons. To explain the comment, we add the description in Methods (Line 17-21, Section 2.1, Page 2)

Reviewer 2 Report
Many thanks to the authors for having presented their cases series. The topic is interesting and has a potential interest for the orthopedic community. The concept of the study is well described, even if there are some aspects that are not immediately clear for readers.
Specific Comment:
- The inclusion criteria are not well described, and I suggest improving this topic.
- Page 3 line 90: “Until now, the National Health Insurance Administration of our country only offers 90 ST screws for patients who need instrumentation. Therefore, patients have to pay at their 91 own expense or private insurance supported for using DT screw. We used both ST and 92 DT screws in patients who underwent CBT surgery based on patients’ insurance situation 93 and agreement of self-paying.” The treatment you present, differs for “rich people” and for “not rich people”. This fact may impact on the results. In other hand, the “not rich people” may have a way of life that increase the risk of screw mobilization such as heavy work, bad general conditions, ecc.
- Page 4 line 131: “Although older age 131 was manifested in DT group (P =0.0055**), 29 out of 73 (39.7%) patients in ST group and 132 only 10 out of 86 (11.6%) patients in DT group exhibited halo sign or screw motion on the 133 plain films (Fisher’s exact test, P < 0.0001****)”. This age difference may affect the results. How did you explain the results?
- An Xray of the halo sign of the series is missing. Would you provide an Xray with halo sign and one without it?
- Page 4 line 134: “Patients in group ST and DT were classified into subgroups ST-1L, ST-2L, and ST-3L and DT-1L, DT-2L, and DT-3L, which represent the fixation lengths of 1, 2, and 3 levels. We found that the incidences of halo phenomenon in subgroups ST-1L, DT-1L, ST-2L, DT-2L, ST-3L, and DT-3L were 3 in 27 (11.1%), 1 in 33 (3%), 10 in 27 (37%), 5 in 36 (13.8%), 16 in 19 (84.2%), and 4 in 17 (23.5%), respectively (Fig. 3), suggesting DT group can reduce the incidence rates of halo phenomenon resulted from the greater of fixation length.” Did you find any statistical difference in the results?
- The results are chaotic and should be better organized.
- There are many typos.
I think that the final message could be misunderstood from the reader.
Author Response
We greatly appreciate the efforts of the Editorial Board and the reviewers for giving us valuable comments on our manuscript. As requested, we enclosed a revised version of our manuscript in response to the extensive and insightful reviewer’s comment. Included below is our point-by-point response to the comment.
1.The inclusion criteria are not well described, and I suggest improving this topic.
As reviewer’s suggestion, we edit the description of inclusion criteria more detail in methods (Line 5-10, Section 2.1, Page 2). Moreover, the aim of this study is to ask if the different type of CBT affect the risk of screw loosening by subgrouping and many aspects of statistically verification, and we think the topic is concise and easy to reader.
2.Page 3 line 90: “Until now, the National Health Insurance Administration of our country only offers 90 ST screws for patients who need instrumentation. Therefore, patients have to pay at their 91 own expense or private insurance supported for using DT screw. We used both ST and 92 DT screws in patients who underwent CBT surgery based on patients’ insurance situation 93 and agreement of self-paying.” à The treatment you present, differs for “rich people” and for “not rich people”. This fact may impact on the results. In other hand, the “not rich people” may have a way of life that increase the risk of screw mobilization such as heavy work, bad general conditions, etc.
Although the fact that insurance differences may discriminate the people and that would be affect the results, we still think that have to be disclosed. Because neither patients nor surgeons realized the difference of fixation strength between ST and DT screw because there were no reports have verified the clinical outcomes in CBT with differently designed thread screws. Moreover, the initial idea we used DT screws is due to the cannulated design, which can reduce the radiation exposure of operative surgeons. To explain the comment, we add the description in Methods (Line 17-21, Section 2.1, Page 2)
3.Page 4 line 131: “Although older age 131 was manifested in DT group (P =0.0055**), 29 out of 73 (39.7%) patients in ST group and 132 only 10 out of 86 (11.6%) patients in DT group exhibited halo sign or screw motion on the 133 plain films (Fisher’s exact test, P < 0.0001****)”. à This age difference may affect the results. How did you explain the results?
Even the older age in DT group, the loosening rate of DT group was lower than ST group. As reviewer’s comment. We add the description in Results (Line 3-5, Paragraph 1, Page 5)
4.A X-ray of the halo sign of the series is missing. Would you provide an Xray with halo sign and one without it?
As reviewer’s suggestion, we present the X-ray of halo sign in Figure 2 (Single and dual).
5.Page 4 line 134: “Patients in group ST and DT were classified into subgroups ST-1L, ST-2L, and ST-3L and DT-1L, DT-2L, and DT-3L, which represent the fixation lengths of 1, 2, and 3 levels. We found that the incidences of halo phenomenon in subgroups ST-1L, DT-1L, ST-2L, DT-2L, ST-3L, and DT-3L were 3 in 27 (11.1%), 1 in 33 (3%), 10 in 27 (37%), 5 in 36 (13.8%), 16 in 19 (84.2%), and 4 in 17 (23.5%), respectively (Fig. 3), suggesting DT group can reduce the incidence rates of halo phenomenon resulted from the greater of fixation length.” à Did you find any statistical difference in the results?
As reviewer’s suggestion, we add the statistical difference in Results (Line 11-13, 19-21, Paragraph 1, Page 5)
6.The results are chaotic and should be better organized.
As reviewer’s suggestion, we have arranged and add statistical difference in Results section (Line 11-13, 19-21, Paragraph 1, Page 5)
7.There are many typos.
As reviewer’s suggestion, we correct some typos and references journal style in the manuscript.
